materials science/supramolecular chemistry/inorganic chemistry

carbon dioxide, metal–organic framework, MOF, manganese, amine modification, solvent recycling

**Authors for correspondence:**
Naseem Iqbal
e-mail: naseem@casen.nust.edu.pk
Timothy L. Easun
e-mail: easuntl@cardiff.ac.uk

This article has been edited by the Royal Society of Chemistry, including the commissioning, peer review process and editorial aspects up to the point of acceptance.

# Ethylenediamine loading into a manganese-based metal–organic framework enhances water stability and carbon dioxide uptake of the framework

Aisha Asghar[1], Naseem Iqbal[1], Leena Aftab[1], Tayyaba Noor[2], Benson M. Kariuki[3], Luke Kidwell[3] and Timothy L. Easun[3]

[1]U.S.-Pakistan Center for Advanced Studies in Energy (USPCAS-E), and [2]School of Chemical and Materials Engineering (SCME), National University of Sciences and Technology (NUST), H-12, Islamabad 44000, Pakistan
[3]School of Chemistry, Cardiff University, Main Building, Park Place, Cardiff CF10 3AT, UK

LK, 0000-0002-3402-2705; TLE, 0000-0002-0713-2642

Metal–organic frameworks (MOFs) based on 2,5-dihydroxyterepthalic acid (DOBDC) as the linker show very high $CO_2$ uptake capacities at low to moderate $CO_2$ pressures; however, these MOFs often require expensive solvent for synthesis and are difficult to regenerate. We have synthesized a Mn-DOBDC MOF and modified it to introduce amine groups into the structure by functionalizing its metal coordination sites with ethylenediamine (EDA). Repeat framework synthesis was then also successfully performed using recycled dimethylformamide (DMF) solvent. Characterization by elemental analysis, FTIR and thermogravimetric studies suggest that EDA molecules are successfully substituting the original metal-bound DMF. This modification not only enhances the material's carbon dioxide sorption capacity, increasing stability to repeated $CO_2$ sorption cycles, but also improves the framework's stability to moisture. Moreover, this is one of the first amine-modified MOFs that can demonstrably be synthesized using recycled solvent, potentially reducing the future costs of production at larger scales.

# 1. Introduction

Among the greenhouse gases highlighted under the Kyoto Protocol, carbon dioxide is a significant contributor to climate change [1]. Significant effort has been made to develop materials that can capture and remove carbon dioxide from process streams and flue gases. These materials include porous adsorbents, which are solid-state alternatives to the well-known amine scrubbing technologies for $CO_2$ capture and sequestration; they have potentially lower energetic costs, greater environmental sustainability and regenerability [2]. Adsorption processes can be operated in either the pressure or temperature swing mode, or in conjunction with membrane systems. Metal–organic frameworks (MOFs) are a class of nanomaterials comprising metal coordination sites bridged by organic ligands [3–5]. These organic/inorganic three-dimensional hybrid networks often have well-defined structures, are commonly highly crystalline and can have very high surface areas. MOF materials with tunable physical and chemical properties have a wide range of applications in gas sorption and separations [6–12], catalysis [13–15] and contaminant removal [16,17]. Substantial research efforts have been made on $CO_2$ gas adsorption applications. However, many MOFs adsorb $CO_2$ by weak physisorption, such that their $CO_2$ selectivity from low-pressure flue gases is very low [18]. Humidity presents a significant challenge for porous adsorbents, as water can hydrolyse and denature the framework materials [19]. Water molecules can disrupt coordination bonds between the organic ligands and the metal centres, resulting in the disintegration of the MOF structure. Increasingly, however, water-stable MOFs are being reported including examples such as UiO-66, MIL-101, NOTT-400 and NOTT-401 [20–22].

Post-synthetic MOF functionalization can impart desirable properties to the MOF materials which were not attainable using direct synthesis. Framework stability enhancement achieved by amine functionalization has been attributed to masking of the coordinately unsaturated hydrophilic metal sites by the linking amino groups, while the alkyl groups such as ethylene bridges in ethylenediamine (EDA) can impart hydrophobicity to the MOF structure, thus minimizing water uptake and consequent dissociation of the coordination bonds [23]. Numerous functionalization methods have been reported, ranging from 'click' chemistry, to linker modification, and to pore impregnation [24]. Encapsulation of active species within MOF networks using impregnation enables MOF morphology preservation and enhanced adsorption applications, especially for carbon dioxide gas capture [25,26].

2,5-Dihydroxyterephthalate linker-based MOFs have been reported to have attractive carbon dioxide capture due to their very high $CO_2$ uptake, favourable structural characteristics, enhanced surface areas over zeolites and ease of synthesis [27]. The large density of unsaturated metal centres and cylindrical pore structure of the MOF-74 family of frameworks provides readily accessible strong binding sites for $CO_2$, but these MOF materials are moisture sensitive [28]. Additionally, while MOF materials based on the 2,5-dihydroxyterephthalic acid (DOBDC) linker have shown exceptionally high $CO_2$ uptake capacities at low to moderate $CO_2$ pressures, these MOFs often require expensive solvent for synthesis and are difficult to regenerate [29]. Even though Mn-DOBDC MOFs have been synthesized in 'green' solvents, including water, there are no reports of re-using the original synthesis solvent to make additional batches of MOF material [30]. We have recently reported the modification of a copper-based MOF during synthesis by doping with hexamethylenetetramine, resulting in the enhancement of carbon dioxide sorption over the unmodified framework [31]. This present study is an attempt to (i) improve on this method by incorporating basic EDA molecules within a new Mn-DOBDC MOF structure to enhance carbon dioxide capture and increase water stability, and (ii) cut the cost for large-scale MOF synthesis by producing the Mn-DOBDC MOF using recycled solvent, demonstrating the potential for routes towards low-cost efficient solid-state amino-MOF adsorbents.

# 2. Material and methods

All the chemicals were purchased from Merck Sigma-Aldrich and used as received.

## 2.1. $\{C_{14}H_{18}MnN_2O_8\}_\infty$ (Mn-DOBDC) synthesis

To prepare Mn-DOBDC, $Mn(NO_3)_2.6H_2O$ (574 mg, 2 mmol) and 2,5-dihydroxyterephthalic acid (396 mg, 2 mmol) were dissolved in 50 ml of dimethylformamide (DMF). The contents were sonicated at 30°C for 20 min then the solution was dispensed into five pressure tubes which were heated at 110°C for 26 h. The solid product was collected by filtration. Crystals obtained thus were washed three times with 10 ml methanol and then three times with 10 ml DMF. The resulting crystals were activated by heating at

**Figure 1.** Reaction scheme for EDA-Mn-DOBDC synthesis.

130°C under dynamic vacuum on a Schlenk line for 14 h. This yielded brown crystals of Mn-DOBDC (89% yield).

## 2.2. Mn-DOBDC post-synthetic amine modification

Synthesized Mn-DOBDC crystals were modified using EDA. About 200 mg of Mn-DOBDC crystals were added to 20% EDA solution in ethanol (5 ml). Contents were heated under reflux with stirring for 8 h. The product obtained was filtered and then washed, first with DI water then with ethanol, to remove any unreacted EDA. The sample was dried for 8 h at room temperature to afford off-white crystals of EDA-Mn-DOBDC. The crystalline product was activated by heating at 140°C under dynamic vacuum on a Schlenk line for 14 h. Synthesis experiments were performed thrice, to ensure reproducibility of results, with an average yield of 76%. The reaction scheme for EDA-Mn-DOBDC is shown in figure 1.

## 2.3. Solvent recovery and recycling

DMF is a commonly used aprotic solvent with a high boiling point. It decomposes at much higher temperatures than those used in this study and can hydrolyse in the presence of acid/base [32]. Pure dry DMF was used and solvent recycling was trialled in this study. All reagents used in these MOFs synthesis are low-cost except the solvent (dimethylformamide; Sigma-Aldrich Corporation online (UK) pricing 31 July 2019 of £61.40 for 1 l, ACS Reagent grade, 99.8%, product code 33120–1 L-M). Fresh DMF was used for batch-I synthesis. Filtrate solvent after crystal collection by sintered filtration was used for batch-II synthesis. The batch-II MOF obtained was characterized using powder X-ray diffraction (PXRD) and compared against synthesized batch-I MOF.

## 2.4. Characterization

Single-crystal X-ray diffraction data for Mn-DOBDC MOF was collected on an Agilent SuperNova Dual Atlas diffractometer with a Mo source and a CCD detector. Data reduction and integration were performed using CrysAlisPro. PXRD patterns were collected on X'PertPro Panalytical Chiller 59 diffractometer using copper $K\alpha$ (1.54 Å) radiation. The $2\theta$ range to record diffraction pattern was from 5 to 40 degrees. A Shimadzu IR Affinitt-1S spectrometer was used to obtain IR spectra. Thermogravimetic analyses (TGA) were performed using a Perkin Elmer Pyris 1 TGA equipment. The temperature was increased from 25°C to 700°C at a heating rate of 5°C min$^{-1}$ under a flow of air (20 ml min$^{-1}$). Elemental analyses were performed using a FlashSmart NC ORG elemental analyser.

CO$_2$ adsorption experiments were performed on a Quantachrome Isorb-HP100 volumetric type sorption analyser. Samples were degassed at 150°C under vacuum for 10 h and then back-filled with helium gas, prior to CO$_2$ sorption studies. Sorption studies were performed at two selected temperatures, 273 and 298 K, over a pressure range of 0.5–15 bar. N$_2$ adsorption studies of prepared samples were conducted to analyse surface area and pore volume using a Quantachrome Nova 2200e at 77 K at a relative pressure of $P/P^O = 0.05$–1.0.

# 3. Results and discussion

## 3.1. Characterization

The Mn-DOBDC MOF crystallizes in a monoclinic cell with $a = 9.6916(5)$ Å, $b = 11.8690(6)$ Å and $c = 15.3430(9)$ Å, and $\alpha = 90°$, $\beta = 102.788(6)°$ and $\gamma = 90°$, with broadly rhomboidal pores occupied by

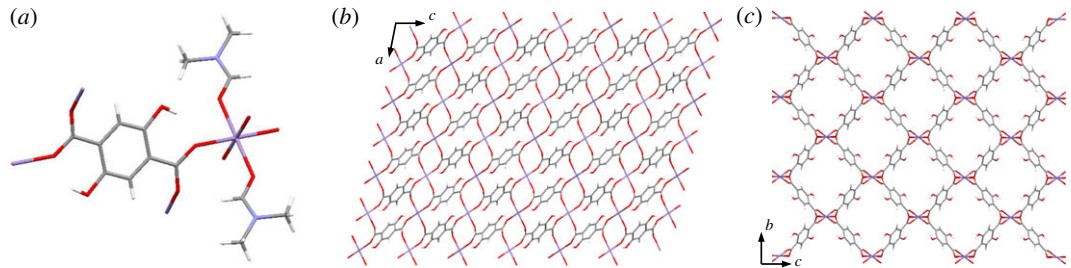

**Figure 2.** Crystal structure of Mn-DOBDC. (*a*) Unit cell showing the Mn (II) and ligand coordination environments; (*b*) view down the *b*-axis of the structure, showing the carboxylate bridging of adjacent Mn (II) ions; (*c*) view down the *a*-axis of the structure showing the rhombohedral channels. DMF molecules are omitted in (*b*) and (*c*) for clarity. Purple atoms represent Mn, red are oxygen, grey are carbon, blue are nitrogen and white are hydrogen atoms.

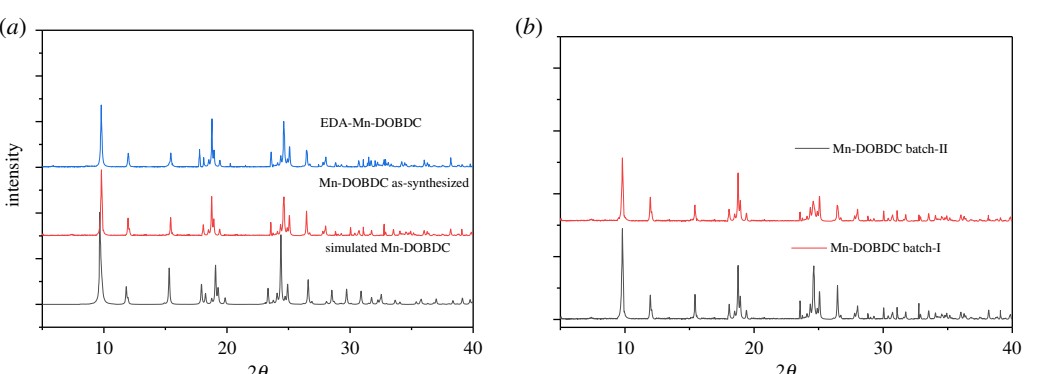

**Figure 3.** (*a*) PXRD patterns for Mn-DOBDC (*black*, simulated from SCXRD), Mn-DOBDC (*red*, synthesized herein), EDA-Mn-DOBDC (*blue*, EDA modified). (*b*) Mn-DOBDC synthesis with fresh DMF (batch-I) and Mn-DOBDC synthesis with recycled DMF (batch-II).

metal-bound DMF solvent molecules. Use of pure DMF as solvent produced this distinct structure compared to related DOBDC linker-based MOFs [23]. The Mn-DOBDC cell comprises one Mn, fourteen carbon atoms, two nitrogen, eight oxygen and eighteen hydrogen atoms. Each Mn atom is octahedrally coordinated with six oxygen atoms; two from axially coordinated DMF and four from carboxylate groups that each bridge two adjacent Mn ions. Further details are given in electronic supplementary material, table S1 and the structure is shown in figure 2.

Electronic supplementary material, figure S1 shows the Fourier transform infrared (FTIR) spectra of unmodified Mn-DOBDC and EDA-modified Mn-DOBDC. FTIR collected for prepared materials confirmed the presence of representative functional groups indicative of Mn-DOBDC MOF formation. Sharp peaks representative of symmetric and asymmetric stretching of carboxylates bonded to Mn are observed at 1535 and 1367 cm$^{-1}$ in Mn-DOBDC sample [33]. Both samples contain a broad band at around 3250 cm$^{-1}$, which can be attributed to O–H stretching vibrations of adsorbed atmospheric water. This broad band centred around 3250 cm$^{-1}$ is notably reduced in the EDA-Mn-DOBDC sample, which is perhaps indicative of slower water adsorption as a result of pore-blocking by adsorbed EDA molecules incorporated into the MOF structure and is consistent with the improved water stability of the modified MOF (see below) [34]. Bands in the 1600 to 800 cm$^{-1}$ region are due to aromatic ring stretching [35,36]. In addition to peaks coincident with Mn-DOBDC sample, the EDA-Mn-DOBDC spectrum has new peaks at 2915 and 2830 cm$^{-1}$ that can be ascribed to stretching vibrations of C–H bonds introduced by the incorporation of EDA molecules [37,38] and bands at 3375, 3280, 2935, 2815 and 1540 cm$^{-1}$ which match well with the FTIR spectrum of EDA, confirming that EDA has been successfully incorporated into the Mn-DOBDC MOF [39,40].

The PXRD patterns of the as-synthesized Mn-DOBDC, the pattern simulated from the single-crystal XRD and the EDA-Mn-DOBDC-modified material are shown in figure 3*a*. The similarity of all three is excellent, indicating the absence of amorphous material, good phase purity and retention of the overall MOF structure on modification with EDA.

The FTIR spectrum of fresh DMF used for the batch-I Mn-DOBDC synthesis closely resembles the FTIR spectrum of the recycled DMF recovered after the collection of the batch-I product (electronic

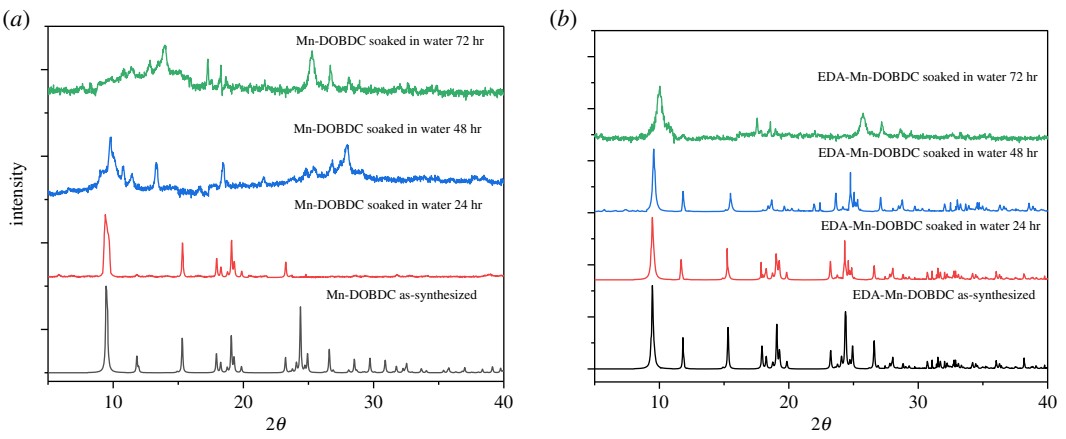

**Figure 4.** (*a*) PXRD patterns for Mn-DOBDC as-synthesized (black), after soaking in water for 24 h (red), 48 h (blue) and after 72 h (green). (*b*) PXRD patterns for EDA-Mn-DOBDC as-synthesized (black), after soaking in water for 24 h (red), 48 h (blue) and after 72 h (green).

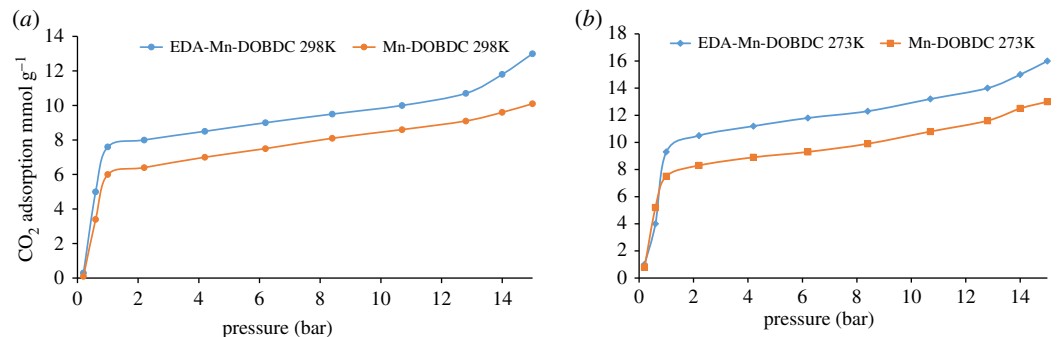

**Figure 5.** $CO_2$ adsorption isotherms in mmol $g^{-1}$ for Mn-DOBDC and EDA-Mn-DOBDC at 273 K (*a*) and 298 K (*b*).

supplementary material, figure S2). The latter solvent was then used to successfully make batch-II of Mn-DOBDC. Figure 3*b* compares the batch-I and batch-II product PXRD patterns, which indicate that the second synthesis produced highly crystalline Mn-DOBDC using the recycled DMF (figure 3*b*). SEM images of batch-I and batch-II products are shown in electronic supplementary material, figure S4. Small changes in some peak intensities observed for these MOF samples may be ascribed to differences in crystallite size between samples, or preferred orientation effects obtained by slight changes in sample preparation [36,38].

Water stability of the MOFs was also explored: the as-synthesized samples of Mn-DOBDC and EDA-Mn-DOBDC were soaked in water at room temperature for three days. PXRD patterns were collected and compared for both materials every 24 h (figure 4*a*,*b*). The corresponding experimental PXRD results indicate that Mn-DOBDC MOF suffers significant loss of crystallinity after 24 h of water exposure while EDA-Mn-DOBDC retains a high degree of crystallinity even after 48 h of soaking, demonstrating the much improved water stability of EDA-Mn-DOBDC.

TGA was performed on both Mn-DOBDC and EDA-Mn-DOBDC (electronic supplementary material, figure S3). For both MOFs, there is no significant weight loss observed below 100°C, indicating there was little surface-adsorbed moisture. In both samples, there is then an initial weight loss step, between approximately 100 and 200°C which we ascribe to the loss of coordinated DMF in Mn-DOBDC and the loss of coordinated EDA in EDA-Mn-DOBDC. This weight loss step occurs at approximately 25°C lower for the EDA-Mn-DOBDC, consistent with the higher volatility of EDA than DMF. For both samples, linker degradation (about 47% in Mn-DOBDC and 51% in EDA-Mn-DOBDC) occurs as a gradual weight loss in both MOFs as temperature increases above approximately 200°C. No further weight losses were observed above 460°C for Mn-DOBDC and 500°C for EDA-Mn-DOBDC.

The SEM images of particles of Mn-DOBDC and EDA-Mn-DOBDC (electronic supplementary material, figure S4) show a range of particle sizes including well-formed hexagonal crystals measuring approximately 2–4 µm that maintain their morphology after amine functionalization.

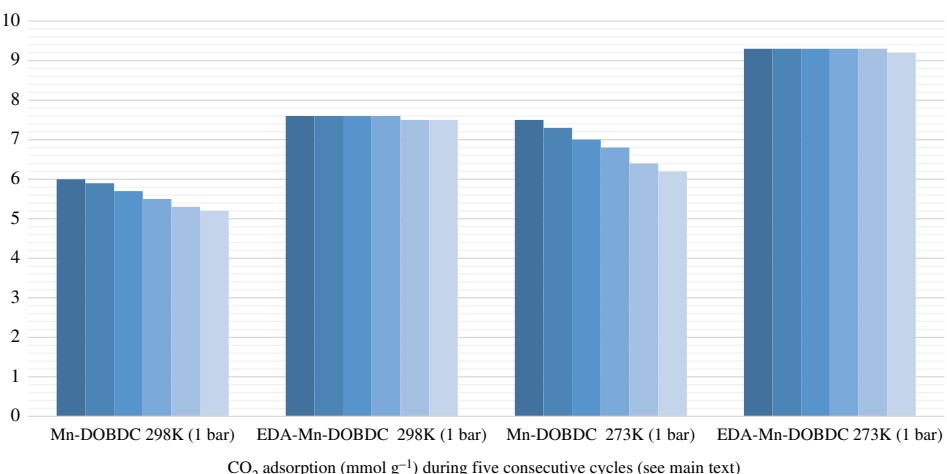

**Figure 6.** $CO_2$ adsorption at 1 bar in mmol $g^{-1}$ for Mn-DOBDC and EDA-Mn-DOBDC at 273 and 298 K.

**Table 1.** Surface area, $CO_2$ uptake and Qst values for selected DOBDC linker-based MOFs.

| material | BET ($m^2/g$) | temperature (K) | pressure (bar) | $CO_2$ adsorption (wt.%) | Qst ($KJmol^{-1}$) | reference |
|---|---|---|---|---|---|---|
| $Fe_2$(DOBDC) | 1345 | 298 | 1 | 30.8 | 33 | [42] |
| | | 308 | | 27.3 | | |
| | | 318 | | 22.9 | | |
| $Mg_2$(DOBDC) | | 313 | 1 | 30.8 | | [43] |
| Mg(DOBDC) EDA | | 298 | 1 | 7.04 | 30 | [44] |
| Mg(DOBDC) | 1525 | 298 | 10 | 37.4 | | [25] |
| | | 313 | | 32.6 | | |
| | | 328 | | 29.9 | | |
| Py-Ni-DOBDC | 409 | 298 | 1 | 12 | | [25] |
| Ni-MOF 74 | 1252 | 298 | 1 | 19.4 | | [45] |
| Mg-MOF-74 | 1416 | 298 | 1 | 30.1 | | [45] |
| Mg-DOBDC | 1415.1 | 298 | 1 | 25 | 47 | [46] |
| Co-DOBDC | 1089.3 | 298 | 1 | 21.6 | 37 | [46] |
| Ni-DOBDC | 1017.5 | 298 | 1 | 20.5 | 42 | [46] |
| Mn-DOBDC | 1256 | 273 | 1 | 33 | 29 | present study |
| | | | 15 | 57.3 | | |
| | | 298 | 1 | 26.4 | | |
| | | | 15 | 44.5 | | |
| EDA-Mn-DOBDC | 1203 | 273 | 1 | 40.9 | 32 | present study |
| | | | 15 | 70.4 | | |
| | | 298 | 1 | 33.5 | | |
| | | | 15 | 57.2 | | |

To confirm the chemical composition of both samples, elemental analyses were performed (electronic supplementary material, table S2). The empirical formulae calculated on the basis of elemental analysis for Mn-DOBDC and EDA-Mn-DOBDC are $C_{14}H_{18}MnN_2O_8$ and $C_{12}H_{22}MnN_4O_6$, respectively, consistent with a metal : linker : DMF ratio of 1 : 1 : 2 for Mn-DOBDC and metal : linker : EDA molar ratio of 1 : 1 : 2

for EDA-Mn-DOBDC. This is consistent with the removal of the axially coordinated DMF molecules (figure 2a) from Mn-DOBDC and replacement with EDA molecules in the amine modification process.

## 3.2. $CO_2$ adsorption capacities of Mn-DOBDC and EDA-Mn-DOBDC

The $CO_2$ adsorption capacity for both MOF materials was evaluated by monitoring pseudo equilibrium adsorption uptakes. Initially, samples were degassed at 150°C for 10 h using a heating rate of 5°C min$^{-1}$. About 200 mg of each sample was used for three consecutive adsorption–desorption cycles at 273 and 298 K with adsorbate pressure ranging between 0.1 and 15 bar. The $CO_2$ capacities calculated at 273 K and 15 bar pressure were 12.5 and 16 mmol g$^{-1}$ for Mn-DOBDC and EDA-Mn-DOBDC, respectively. This trend also occurs for adsorption capacities recorded at 298 K (figure 5a). Here we observed that $CO_2$ uptake for Mn-DOBDC was higher for initial adsorption cycles at both 273 and 298 K, but adsorption capacity declined considerably under successive cycles (figure 6). This fact perhaps implies that Mn-DOBDC demands more energy input for its regeneration than applied in this work, possibly making a $CO_2$ capture process with this adsorbent significantly more energy intensive [41]. By contrast, EDA-Mn-DOBDC indicated complete regenerability under the desorption conditions used here, showing negligible decline in $CO_2$ adsorption capacity over six successive test cycles. The $N_2$ adsorption isotherm for the MOFs was recorded at 77 K (electronic supplementary material, figure S5). The Langmuir and BET surface areas for Mn-DOBDC MOF were found to be 1583 and 1256 m$^2$ g$^{-1}$, respectively, while EDA-Mn-DOBDC revealed lower values of 1415 m$^2$ g$^{-1}$ (Langmuir) and 1203 m$^2$ g$^{-1}$ (BET) (table 1). Although the introduction of the amine into EDA-Mn-DOBDC slightly reduces MOF surface area, $CO_2$ adsorption noticeably increased compared to various other DOBDC linker-based MOF materials (table 1). The presence of additional binding sites in MOFs by amine/amide incorporation has been shown to induce dispersion and electrostatic forces that enhance $CO_2$ gas adsorption [47]. Isosteric heats of adsorption for Mn-DOBDC and EDA-Mn-DOBDC were calculated from isotherms recorded at 273 and 298 K (see electronic supplementary material, S1 and Figure S6). Qst at zero loading is approximately 3 kJ mol$^{-1}$ higher for the amine-modified framework.

## 4. Conclusion

Competitive carbon dioxide adsorbent-based technologies need sorbents with high gas uptake capacities and low production cost. Low-cost amine-doping of Mn-DOBDC has produced EDA-Mn-DOBDC, which demonstrates high $CO_2$ uptake capacity and enhanced stability to water. The addition of nitrogen atoms by the incorporation of EDA molecules leads to the enhanced adsorption of $CO_2$ gas, which we ascribe to favourable interactions between $CO_2$ molecules and the nitrogen-modified pores [48]. Stability enhancement as achieved by amine functionalization may be partially attributed also to masking of the co-ordinately unsaturated hydrophilic metal sites by the linking amino groups, thus minimizing water absorption by the MOF, hence minimizing competition between water and $CO_2$ at the amine binding sites [23]. The moderate isosteric heat of $CO_2$ adsorption (Qst) for EDA-Mn-DOBDC is highly desirable because of the anticipated lower regeneration energy demand. $CO_2$ uptake was good, and, significantly, almost negligible $CO_2$ uptake loss was observed over six consecutive adsorption cycles for EDA-Mn-DOBDC, while only relatively mild regeneration conditions were required. We were also able to re-use the original synthesis solvent to successfully make a new batch of the MOF, which is a very promising strategy to improve the cost effectiveness of MOF synthesis. Finally, the improvement in framework stability to water exposure after modification with EDA makes this strategy potentially applicable in real-world carbon dioxide capture applications in the future.

Data accessibility. The crystallographic dataset supporting this article has been uploaded in the electronic supplementary material and has been submitted to the Cambridge Crystallographic Data Centre, reference CCDC 1949695. These data can be obtained free of charge via https://www.ccdc.cam.ac.uk/structures/.

Authors' contributions. N.I. and T.N. conceptualized this project, A.A. carried out MOF synthesis and characterization. N.I. and T.L.E. analysed results. B.M.K. and L.K. collected single-crystal data and solved the crystal structure. A.A., L.A. and T.L.E. prepared the manuscript, T.L.E. and N.I. supervised the overall design, synthesis and development of the project.

Competing interests. We declare we have no competing interests.

Funding. The authors are grateful to the United States Centre for Advanced Studies in Energy, Pakistan and Higher Education Commission, Pakistan for providing financial support under NRPU programme (project no. 6013) and IRSIP programme to carry out this research work at Cardiff University, UK. T.L.E. gratefully acknowledges the Royal Society for the award of a University Research Fellowship (6866) and Cardiff University for funding.

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
