## [Reviewer comments · Royal Society Open Science]

Review History

RSOS-191934.R0 (Original submission)

Review form: Reviewer 1

Is the manuscript scientifically sound in its present form?

Yes

Are the interpretations and conclusions justified by the results?

Yes

Is the language acceptable?

Yes

Do you have any ethical concerns with this paper?

No

Have you any concerns about statistical analyses in this paper?

No

Recommendation?

Accept with minor revision (please list in comments)

Comments to the Author(s)

The contribution by Iqbal, Easun and co-workers entitled "Ethylenediamine (EDA) loading into a manganese-based MOF enhances water stability and carbon dioxide uptake of the framework" demonstrates an interesting strategy to enhance water stability of the MOF material and augment its CO₂ capture capacity. In general this contribution is well presented and the results are interesting. This paper should be accepted after only minor corrections.

1) In the introduction it should be added a small paragraph about MOFs stable to water. Some literature to cite: *Materials Chemistry Frontiers* 2017, 1, 1471-1484; *Inorganic Chemistry Frontiers* 2015, 2, 1080-1084; *Inorganic Chemistry Frontiers* 2015, 2, 442-447.

2) Figure 4a should be improved.

3) How can the authors explain the increase on the CO₂ capture when loading EDA? Can the authors provide a mechanism? This reference should be cited and it also can be helpful to provide a hypothesis for this CO₂ enhancement: *Inorganic Chemistry* 2017, 56, 5863-5872.

Review form: Reviewer 2

Is the manuscript scientifically sound in its present form?

No

Are the interpretations and conclusions justified by the results?

No

Is the language acceptable?

Yes

Do you have any ethical concerns with this paper?

No

Have you any concerns about statistical analyses in this paper?

No

Recommendation?

Major revision is needed (please make suggestions in comments)

Comments to the Author(s)

In this manuscript, the authors report an ethylene diamine modified Mn-DOBDC MOF for CO₂ uptake. In addition, the authors also demonstrate that the DMF solvent that was used for MOF preparation was recycled and reused for the same synthesis. Overall, the findings in this work are interesting and enrich the knowledge about amine-modified MOFs for CO₂ uptake. One of the highlights for this paper is the reuse of the DMF solvent because the amine modification on MOF has been commonly reported. However, some conclusions related to that are not well supported by the data and need further investigation and discussion. I would recommend a major revision before acceptance.

1. The authors recycled and reused the DMF solvent. However, no characterizations on the recycled DMF were shown in the manuscript. Are there any changes to the recycled DMF? I believe FTIR spectroscopy could tell that.

2. In Figure 3, the XRD patterns for the Mn-DOBDC prepared from the pristine and the recycled DMF exhibit different peak intensities, where batch-II showed weaker signals. In addition, the peaks at around 25 and 26 are much weaker. What are the possible reasons for those differences?
3. As there are some differences in the XRD patterns, is the morphology of batch-II also altered?
4. What is the stability of batch-II against water? Is it similar to the batch-I?
5. Could the DMF be recycled further, e.g. three, four, and even more times?

Review form: Reviewer 3

Is the manuscript scientifically sound in its present form?

Yes

Are the interpretations and conclusions justified by the results?

Yes

Is the language acceptable?

Yes

Do you have any ethical concerns with this paper?

No

Have you any concerns about statistical analyses in this paper?

No

Recommendation?

Major revision is needed (please make suggestions in comments)

Comments to the Author(s)

This work reports a Mn-MOF loaded with ethylenediamine for the enhancement of CO₂ uptake and water stability of material. However, the significance of this work is not enough for the publication in Royal Society Open Science. Several issues need to be addressed:

1. DMF is known to decompose under high temperature, the recycled DMF solution should contain different components. How many times can the recycled solvent be repeatedly used under the same reaction conditions?
2. The authors mentioned about synthesizing MOFs using recycled solvent. Why only the PXRD patterns were showed, no other characterization?
3. The PXRD patterns in Figure 3A are apparently different by peak positions and numbers. Please explain it.
4. Amine-modified MOF for CO₂ adsorption has been studied for a long time. Please provide a comparison why the results are significant. Especially the structure of this MOF is so distinct from all other DOBDC-based MOFs listed in Table 1.

Decision letter (RSOS-191934.R0)

10-Dec-2019

Dear Dr Easun:

Title: Ethylenediamine (EDA) loading into a manganese-based MOF enhances water stability and carbon dioxide uptake of the framework.

Manuscript ID: RSOS-191934

The editor assigned to your manuscript has now received comments from reviewers. We would like you to revise your paper in accordance with the referee and Subject Editor suggestions which can be found below (not including confidential reports to the Editor). Please note this decision does not guarantee eventual acceptance.

Please submit your revised paper before 02-Jan-2020. Please note that the revision deadline will expire at 00.00am on this date. If we do not hear from you within this time then it will be assumed that the paper has been withdrawn. In exceptional circumstances, extensions may be possible if agreed with the Editorial Office in advance. We do not allow multiple rounds of revision so we urge you to make every effort to fully address all of the comments at this stage. If deemed necessary by the Editors, your manuscript will be sent back to one or more of the original reviewers for assessment. If the original reviewers are not available we may invite new reviewers.

RSC Associate Editor:
Comments to the Author:
(There are no comments.)

RSC Subject Editor:
Comments to the Author:
(There are no comments.)

Reviewers' Comments to Author:

Reviewer: 1

Comments to the Author(s)

The contribution by Iqbal, Easun and co-workers entitled "Ethylenediamine (EDA) loading into a manganese-based MOF enhances water stability and carbon dioxide uptake of the framework" demonstrates an interesting strategy to enhance water stability of the MOF material and augment its CO₂ capture capacity. In general this contribution is well presented and the results are interesting. This paper should be accepted after only minor corrections.

1) In the introduction it should be added a small paragraph about MOFs stable to water. Some literature to be cited: *Materials Chemistry Frontiers* 2017, 1, 1471-1484; *Inorganic Chemistry Frontiers* 2015, 2, 1080-1084; *Inorganic Chemistry Frontiers* 2015, 2, 442-447.

2) Figure 4a should be improved.

3) How can the authors explain the increase on the CO₂ capture when loading EDA? Can the authors provide a mechanism? This reference should be cited and it also can be helpful to provide a hypothesis for this CO₂ enhancement: *Inorganic Chemistry* 2017, 56, 5863-5872.

Reviewer: 2

Comments to the Author(s)

In this manuscript, the authors report an ethylene diamine modified Mn-DOBDC MOF for CO₂ uptake. In addition, the authors also demonstrate that the DMF solvent that was used for MOF preparation was recycled and reused for the same synthesis. Overall, the findings in this work are interesting and enrich the knowledge about amine-modified MOFs for CO₂ uptake. One of the highlights for this paper is the reuse of the DMF solvent because the amine modification on MOF has been commonly reported. However, some conclusions related to that are not well supported by the data and need further investigation and discussion. I would recommend a major revision before acceptance.

1. The authors recycled and reused the DMF solvent. However, no characterizations on the recycled DMF were shown in the manuscript. Are there any changes to the recycled DMF? I believe FTIR spectroscopy could tell that.
2. In Figure 3, the XRD patterns for the Mn-DOBDC prepared from the pristine and the recycled DMF exhibit different peak intensities, where batch-II showed weaker signals. In addition, the peaks at around 25 and 26 are much weaker. What are the possible reasons for those differences?
3. As there are some differences in the XRD patterns, is the morphology of batch-II also altered?
4. What is the stability of batch-II against water? Is it similar to the batch-I?
5. Could the DMF be recycled further, e.g. three, four, and even more times?

Reviewer: 3

Comments to the Author(s)

This work reports a Mn-MOF loaded with ethylenediamine for the enhancement of CO₂ uptake and water stability of material. However, the significance of this work is not enough for the publication in Royal Society Open Science. Several issues need to be addressed:

1. DMF is known to decompose under high temperature, the recycled DMF solution should contain different components. How many times can the recycled solvent be repeatedly used under the same reaction conditions?
2. The authors mentioned about synthesizing MOFs using recycled solvent. Why only the PXRD patterns were showed, no other characterization?
3. The PXRD patterns in Figure 3A are apparently different by peak positions and numbers. Please explain it.

4. Amine-modified MOF for CO₂ adsorption has been studied for a long time. Please provide a comparison why the results are significant. Especially the structure of this MOF is so distinct from all other DOBDC-based MOFs listed in Table 1.

Author's Response to Decision Letter for (RSOS-191934.R0)

See Appendix A.

RSOS-191934.R1 (Revision)

Review form: Reviewer 2

Is the manuscript scientifically sound in its present form?

Yes

Are the interpretations and conclusions justified by the results?

Yes

Is the language acceptable?

Yes

Do you have any ethical concerns with this paper?

No

Have you any concerns about statistical analyses in this paper?

No

Recommendation?

Accept with minor revision (please list in comments)

Comments to the Author(s)

The authors have addressed most of the reviewers' issues. Regarding the 3rd question from Reviewer 2, it is better to show the SEM image (maybe in the supporting information) for batch-II to compare with that of batch-I.

Decision letter (RSOS-191934.R1)

30-Jan-2020

Dear Dr Easun:

Title: Ethylenediamine (EDA) loading into a manganese-based MOF enhances water stability and carbon dioxide uptake of the framework.

Manuscript ID: RSOS-191934.R1

Thank you for submitting the above manuscript to Royal Society Open Science. On behalf of the Editors and the Royal Society of Chemistry, I am pleased to inform you that your manuscript will

be accepted for publication in Royal Society Open Science subject to minor revision in accordance with the referee suggestions. Please find the reviewers' comments at the end of this email.

The reviewers and handling editors have recommended publication, but also suggest some minor revisions to your manuscript. Therefore, I invite you to respond to the comments and revise your manuscript.

Because the schedule for publication is very tight, it is a condition of publication that you submit the revised version of your manuscript before 08-Feb-2020. Please note that the revision deadline will expire at 00.00am on this date. If you do not think you will be able to meet this date please let me know immediately.

Best wishes,
Dr Laura Smith
Publishing Editor, Journals

RSC Associate Editor:
Comments to the Author:
(There are no comments.)

RSC Subject Editor:
Comments to the Author:
(There are no comments.)

Reviewer comments to Author:
Reviewer: 2

Comments to the Author(s)
The authors have addressed most of the reviewers' issues. Regarding the 3rd question from Reviewer 2, it is better to show the SEM image (maybe in the supporting information) for batch-II to compare with that of batch-I.

Author's Response to Decision Letter for (RSOS-191934.R1)

See Appendix B.

Decision letter (RSOS-191934.R2)

24-Feb-2020

Dear Dr Easun:

Title: Ethylenediamine (EDA) loading into a manganese-based MOF enhances water stability and carbon dioxide uptake of the framework.
Manuscript ID: RSOS-191934.R2

It is a pleasure to accept your manuscript in its current form for publication in Royal Society Open Science. The chemistry content of Royal Society Open Science is published in collaboration with the Royal Society of Chemistry.

RSC Associate Editor
Comments to the Author:
(There are no comments.)

Reviewer(s)' Comments to Author:

Appendix A

Dr. Laura Smith
Publishing Editor,
Royal Society Open Science

Dear Dr Smith,

Re: RSOS-191934

Please find attached the revised manuscript "**Ethylenediamine (EDA) loading into a manganese-based MOF enhances water stability and carbon dioxide uptake of the framework**" for publication as a paper in Royal Society Open Science. We were pleased with the reviewers' positive comments and have endeavoured to address their requested revisions (detailed further below). We have highlighted the changes made in the main manuscript and SI in **yellow** for ease of reference (pdf files). We have also uploaded 'clean' unhighlighted versions as .doc files.

Thank you for reconsidering the manuscript and we hope that it will now be acceptable for publication.

With all best wishes and thanks,

Dr Timothy L. Easun
on behalf of all authors.

Reviewers' Comments to Authors and Authors' responses:

Reviewer comments are inset with respect to our responses.

Reviewer: 1

The contribution by Iqbal, Easun and co-workers entitled "Ethylenediamine (EDA) loading into a manganese-based MOF enhances water stability and carbon dioxide uptake of the framework" demonstrates an interesting strategy to enhance water stability of the MOF material and augment its CO₂ capture capacity. In general, this contribution is well presented and the results are interesting. This paper should be accepted after only minor corrections.

We thank the reviewer for their careful reading of the paper and helpful comments, which are addressed as follows:

1) In the introduction it should be added a small paragraph about MOFs stable to water. Some literature to cite: *Materials Chemistry Frontiers* 2017, 1, 1471-1484; *Inorganic Chemistry Frontiers* 2015, 2, 1080-1084; *Inorganic Chemistry Frontiers* 2015, 2, 442-447.

Thank you for suggesting these relevant and appropriate publications. We have added the following additional text and incorporated these literature references accordingly:

“Humidity presents a significant challenge for porous adsorbents, as water can hydrolyse and denature the framework materials [19]. Water molecules can disrupt coordination bonds between the organic ligands and the metal centres, resulting in the disintegration of the MOF structure. Increasingly, however, water-stable MOFs are being reported including examples such as UiO-66, MIL-101, NOTT-400, and NOTT-401 [20-22].

Post-synthetic MOF functionalization can impart desirable properties to the MOF materials which were not attainable using direct synthesis. Framework stability enhancement achieved by amine-functionalization has been attributed to masking of the coordinately unsaturated hydrophilic metal sites by the linking amino groups, while the alkyl groups such as ethylene bridges in EDA can impart hydrophobicity to the MOF structure, thus minimizing water uptake and consequent dissociation of the coordination bonds [23].”

2) Figure 4a should be improved.

We agree with this and thank the reviewer; an improved Figure 4a has been included in the revised manuscript. Specific improvements include improved resolution of the figure and higher quality text labels.

3) How can the authors explain the increase on the CO₂ capture when loading EDA? Can the authors provide a mechanism? This reference should be cited and it also can be helpful to provide a hypothesis for this CO₂ enhancement: *Inorganic Chemistry* 2017, 56, 5863-5872.

Thank you for the helpful suggested literature. While (as Reviewer 3 notes) the use of amines to increase CO₂ uptake in porous materials is a well-established phenomenon, the mechanism is

still an area of study and is not definitively assigned in this case. However, we have added the usefully proposed references [50] and [23], which have enabled us to tentatively speculate on the mechanistic reasons for the enhancement in CO₂ uptake. Specifically, we have added the following text in the Conclusion section:

“The addition of nitrogen atoms by the incorporation of EDA molecules leads to the enhanced adsorption of CO₂ gas, which we ascribe to favourable interactions between CO₂ molecules and the nitrogen-modified pores [50]. Stability enhancement as achieved by amine-functionalization may be partially attributed also to masking of the co-ordinately unsaturated hydrophilic metal sites by the linking amino groups, thus minimizing water absorption by the MOF, hence minimising competition between water and CO₂ at the amine binding sites [23].”

Reviewer: 2

In this manuscript, the authors report an ethylene diamine modified Mn-DOBDC MOF for CO₂ uptake. In addition, the authors also demonstrate that the DMF solvent that was used for MOF preparation was recycled and reused for the same synthesis. Overall, the findings in this work are interesting and enrich the knowledge about amine-modified MOFs for CO₂ uptake. One of the highlights for this paper is the reuse of the DMF solvent because the amine modification on MOF has been commonly reported. However, some conclusions related to that are not well supported by the data and need further investigation and discussion. I would recommend a major revision before acceptance.

We greatly thank the reviewer for their positive assessment of our work and for highlighting various issues in the manuscript. We have used these to improve the manuscript, addressing each as follows:

1. The authors recycled and reused the DMF solvent. However, no characterizations on the recycled DMF were shown in the manuscript. Are there any changes to the recycled DMF? I believe FTIR spectroscopy could tell that.

We thank the reviewer for noticing this issue. FTIR spectra for fresh and recycled DMF have been added to the supporting information of the revised manuscript (new Figure S2). The reviewer is right that this analysis is of value; no marked changes were observed in the spectra of fresh and recycled DMF, which supports the recycling approach and helps explain why the recycled synthesis produces the same MOF product. The following text has accordingly been added to Section 3.1:

“The FTIR spectrum of fresh DMF used for the batch-I Mn-DOBDC synthesis closely resembles the FTIR spectrum of the recycled DMF recovered after collection of the batch-I product (Figure S2). The latter solvent was then used to successfully make batch-II of Mn-DOBDC.”

2. In Figure 3, the XRD patterns for the Mn-DOBDC prepared from the pristine and the recycled DMF exhibit different peak intensities, where batch-II showed weaker signals. In addition, the peaks at around 25 and 26 are much weaker. What are the possible reasons for those differences?

The reviewer has keen eyes: while the PXRD patterns are very similar, there are indeed small differences in peak intensities. As we are sure the reviewer is aware, the bulk intensity difference between the two patterns is likely to simply be due to a different quantity of material being present on the PXRD plates for each sample. This is purely experimental, can not readily be controlled for in the setup we have access to, and has no bearing on the scientific conclusions. We note that the key issue of product crystallinity is also not affected by this – there is no baseline removal in the data in either pattern that could mask the presence of differing amounts of amorphous material.

The differences in specific peak intensities may be due to an element of preferred orientation that differs between the two samples as-loaded on the plates for the measurement. The sample manifests as a mixture of small microcrystalline particles and larger ~2-4 μm hexagonal crystals (see SEM in SI), the latter of which may be responsible for preferred orientation effects causing the observed differences.

We have added the following text to the manuscript in Section 3.1:

“Small changes in some peak intensities observed for these MOF samples may be ascribed to differences in crystallite size between samples, or preferred orientation effects obtained by slight changes in sample preparation [37, 39].”

3. As there are some differences in the XRD patterns, is the morphology of batch-II also altered?

SEM and visual inspection shows no readily observed differences in sample morphology between batch-I and batch-II, which is in keeping with the minor nature of the differences in the PXRD patterns.

4. What is the stability of batch-II against water? Is it similar to the batch-I?

Batch-I Mn-DOBDC was found unstable to the moisture exposure so batch-II was not tested. Only EDA-Mn-DOBDC was found stable to the water.

5. Could the DMF be recycled further, e.g. three, four, and even more times?

This is an interesting question that we are addressing in future work and we thank the reviewer for their introductory comment about the solvent recycling being a highlight of the study. Given the success of the first use of recycled solvent and the evidently largely unaffected nature of the solvent as evidenced by the FTIR spectra wisely proposed by the reviewer, we are confident that the recycling approach can work in many-cycle experiments.

The primary focus of this manuscript, however, is the simple modification of a MOF with EDA to improve both water stability and CO_2 uptake of the framework under study, and the recycling of the DMF solvent in that context is a positive addition rather than the key point of this study. However, we are interested in both the general applicability of the solvent recycling approach

and in the specifics of how many cycles could be performed, so work in this area is ongoing and will be published separately as part of a larger work in due course.

Reviewer: 3

This work reports a Mn-MOF loaded with ethylenediamine for the enhancement of CO₂ uptake and water stability of material. However, the significance of this work is not enough for the publication in Royal Society Open Science. Several issues need to be addressed:

We thank the reviewer for highlighting areas where this manuscript can be improved. We note that both Reviewers 1 & 2 highlighted the meaningful contribution that this research article makes to the scientific literature and respectfully disagree with the reviewer's comment on the significance of the work. The issues raised have been addressed as follows:

1. DMF is known to decompose under high temperature, the recycled DMF solution should contain different components. How many times can the recycled solvent be repeatedly used under the same reaction conditions?

The reviewer is correct that DMF is well known to decompose slightly at temperatures around its normal boiling point (153°C) into small quantities of dimethylamine and carbon monoxide (see Comins, Daniel L.; Joseph, Sajan P., 2001, "N,N-Dimethylformamide". *Encyclopedia of Reagents for Organic Synthesis*. John Wiley & Sons. doi:10.1002/047084289x.rd335). It is also true that much higher temperatures (~350°C) are required for significant decomposition to spontaneously occur (see Perrin, D. D.; Armarego, W. L. F. 1988. *Purification of Laboratory Chemicals 3rd ed.*; Pergamon: Oxford., doi.org/10.1002/aheh.19890170605), although the presence of strong acids/bases will accelerate this process.

Our reactions were carried out at 110°C, a temperature significantly below both the decomposition and boiling points of this solvent and hence we did not anticipate problems with solvent decomposition. The risk of hydrolysis was also minimized by using dry DMF. In our work we have not seen evidence of significant decomposition of the solvent. Indeed, as the FTIR spectra recorded in response to point 1 of Reviewer 2 has shown, there are no clear differences between the fresh and recycled DMF in this study. As to the final question in this reviewer's comment, we refer you to our answer to point 5 of Reviewer 2 above, and will ensure we continue to monitor for the decomposition products of DMF in all our future studies on solvent recycling. We thank the reviewer for this insight.

We have added the following text to the manuscript:

“DMF is a commonly used aprotic solvent with a high boiling point. It decomposes at much higher temperatures than those used in this study and can hydrolyse in the presence of acid/base [32]. Pure dry DMF was used and solvent recycling was trialled in this study.”

2. The authors mentioned about synthesizing MOFs using recycled solvent. Why only the PXRD patterns were showed, no other characterization?

The primary focus of this manuscript is the simple modification of a MOF with EDA to improve both water stability and CO₂ uptake of the framework under study, and the recycling of the DMF solvent in that context is a positive addition rather than the key point of this study. The product of the recycled solvent MOF synthesis was characterized by PXRD which is sufficient to show a) the formation of the same bulk product as in the initial synthesis and b) the absence of new amorphous or crystalline products. SEM images of the product also shows a similar outcome to the original synthesis, and the FTIR spectra of the fresh and recycled DMF have been added to the SI. This proof of concept aspect of our work will be expanded on in future studies that are beyond the scope of this manuscript.

3. The PXRD patterns in Figure 3A are apparently different by peak positions and numbers. Please explain it.

It is difficult to know from this comment exactly what the reviewer is referring to. If it is the difference in peak positions and the number of peaks between the black pattern and the other two patterns then this is simply because the black trace is the *simulated* PXRD pattern. This is generated from the X-ray crystal structure as a way to evidence that the bulk crystalline material is indeed that observed in the single-crystal analysis. Since the single-crystal structure is collected at lower temperature than the PXRD measurement, there are crystal compression/expansion effects of temperature that can manifest as peak position shifts along the x-axis when the two data types are compared. Similarly, the zero-position of the powder X-ray diffractometer may be slightly off true-zero, affording a very small misalignment of the simulated and experimental patterns. Either way, these are common effects that are not particularly remarkable in this case.

4. Amine-modified MOF for CO₂ adsorption has been studied for a long time. Please provide a comparison why the results are significant. Especially the structure of this MOF is so distinct from all other DOBDC-based MOFs listed in Table 1.

The significance of these results are partly *because* the MOF is a new MOF that is distinct from the common MOF-74 structural motif, although as referenced in the manuscript other non-MOF-74 MOF structures that contain this ligand are known. In our structure, the coordinated DMF molecules are readily directly substituted for EDA molecules. The ease of modification, the increase in water stability by modification and the increased CO₂ uptake are all important outcomes of the work and, as Reviewer 2 commented, we too believe that “the findings in this work are interesting and enrich the knowledge about amine-modified MOFs for CO₂ uptake”.

We have added a referenced sentence to the crystal structure description section to highlight the distinctive nature of the new structure we have obtained:

“Use of pure DMF as solvent produced this distinct structure compared to related DOBDC linker based MOFs [23].”

Appendix B

Dr. Laura Smith
Publishing Editor,
Royal Society Open Science

Dear Dr Smith,

Re: RSOS-191934.R1

Please find attached the revised manuscript "**Ethylenediamine (EDA) loading into a manganese-based MOF enhances water stability and carbon dioxide uptake of the framework**" for publication as a paper in Royal Society Open Science. We were pleased that the revisions met with the reviewers' satisfaction. The final amendment as requested is that Figure S4 in the supporting information has been updated to include additional SEM images.

Thank you for reconsidering the manuscript and we hope that it will now be acceptable for publication.

With all best wishes and thanks,

Dr Timothy L. Easun
on behalf of all authors.